# Hyoid Bone Syndrome in a Patient Undergoing Left Ventricular Assist Device Implantation

**DOI:** 10.3390/healthcare11081130

**Published:** 2023-04-14

**Authors:** Bruno Bordoni, Allan R. Escher

**Affiliations:** 1Department of Cardiology, Foundation Don Carlo Gnocchi IRCCS, Institute of Hospitalization and Care, S Maria Nascente, Via Capecelatro 66, 20100 Milan, Italy; 2Anesthesiology/Pain Medicine, H. Lee Moffitt Cancer Center and Research Institute, 12902 USF Magnolia Drive, Tampa, FL 33612, USA; allan.escher@moffitt.org

**Keywords:** osteopathic manipulative treatment, osteopathy, fascia, pain, tongue, hyoid bone syndrome

## Abstract

The clinical case describes the presence of hyoid bone syndrome (HBS) in a patient with a left ventricular assist device (LVAD) implantation, and the resolution of painful symptoms through an osteopathic manual technique (unwinding) applied to the tongue. To the knowledge of the authors, it is the first case report involving an LVAD patient with HBS treated with an osteopathic approach. The article briefly reviews the data relating to surgical therapy for patients with a clinical history of end-stage heart failure and symptoms related to HBS and posits some hypotheses on the presence of pain radiating from the hyoid bone to other areas of the body. The text reminds us to place greater clinical emphasis on the palpatory evaluation of the hyoid in the presence of non-specific painful symptoms.

## 1. Introduction

Placement of a left ventricular assist device (LVAD) is a surgical therapy performed for patients with a clinical history of end-stage heart failure, when heart disease does not respond optimally and satisfactorily to drug therapy [1]. The LVAD implant has two main objectives: enable the patient to receive a transplant, by extending the time and possibility of receiving a heart transplant or bridge therapy; or extend the life of a patient, who is unable to receive a transplant or destination therapy, as much as possible [1]. According to the Society of Thoracic Surgeons Interagency Registry for Mechanically Assisted Circulatory Support 2020 Annual Report, and the European Registry for Patients with Mechanical Circulatory Support, over thirty thousand LVADs have been implanted since 2010 [2]. The survival rates one and two years after implantation are around 82.3% and 73.1%, respectively [2]. The HeartMate 3 (HM3, Abbot, Chicago, IL, USA) is currently the most implanted model in the world [2].

The HM3 replaces the work of the left ventricle. It is a magnetically levitated centrifugal-flow pump at the level of the apex of the left ventricle (outside of the pericardium) and, through a 20 mm inflow conduit inside the ventricle, translates about 3–6 L of blood per minute in resting conditions, with a constant flow [3,4]. The pump structure has been designed to reduce the risk of thrombotic formation. The blood is expelled and pushed by the pump towards the ascending aorta, passing through an outflow graft. The pump is powered by external batteries (average duration each of about 17 h), which are connected by a cable that comes out of the abdomen or percutaneous driveline; the same cable sends signals to a controller for reading the parameters of the HM3 [4].

The HM3 implant presents some complications over time. About 30% of LVAD patients suffer from bleeding from the gastrointestinal tract, likely due to arteriovenous morphological changes in the gastric system, and a non-physiological alteration of the von Willebrand factor [2]. Infections resulting from the driveline affect approximately 15–47% of patients [5,6]. Other dangerous events, involving about 11% of patients with an LVAD, although with a lower percentage of rehospitalization with the HM3, are linked to neurological ischemias, the causes of which may derive from machinery malfunctions; thrombus formations; and cardiac arrhythmias [7,8].

Other disorders not always taken into consideration or clinically well-framed in patients with an LVAD, are the problems related to the function of the oropharyngeal area. Likely, one of the most important reasons for this lack of attention and information in the literature is due to the fact that the main research studies are carried out by cardiologists and hardly by other specialists. We know that, after cardiac surgery, many patients suffer from disorders resulting from a persistent or transient functional decline of the oropharynx, the dysfunction of which involves the presence of dysphagia (73.7%), language alterations (not from central neurological lesions, 16.5%), and sleep apnea syndromes (15.6–35.9%) [9,10]. In the literature, only two cases are reported for patients implanted with the HM3, where the presence of dysphagia is described, but for causes related to hemolysis and a consequent dystonia of the smooth muscle of the gastrointestinal tract [11,12]. Recent research highlights that 24% of patients with an LVAD can suffer from sleep apnea syndrome (a percentage resulting from a total of 50 patients), indicating a problem in the oropharyngeal area [13].

This case report describes the resolution of an oropharyngeal problem, caused by the presence of hyoid bone syndrome (HBS) in a patient with an LVAD, through osteopathic manipulative medicine (OMM) of the tongue. To the knowledge of the authors, this is the first clinical case in the literature where HBS, OMM, and a patient with the HM3 are highlighted.

## 2. Case Description

A 48-year-old patient with a previous history of chronic heart failure, coming from another hospital location where he had the LVAD implanted (bridge to transplant), as well as a subcutaneous implantable cardioverter defibrillator (S-ICD, Biotronik AG, Germany), was admitted to the rehabilitation cardiopneumology department. The patient is undergoing antibiotic therapy (doxycycline, 100 mg 2 tablets, Pfizer) for positive Staphylococcus Epidermidis and Staphylococcus Lugdunensis. Ultrasound examination shows a severe reduction in global systolic function (ejection fraction of about 20–25%), no obstruction of the inflow cannula, a normal-sized right ventricle, tricuspid aortic valve with minimal but constant systolic opening of the semilunaris and an absence of significant valve regurgitation. The implantation of the HM3 and S-ICD took place over a month before admission for a cardiovascular rehabilitation cycle.

The patient’s drug therapy was organized as follows: nebivolol 5 mg 1 tablet; omeprazole 20 mg 1 tablet; sacubitril/valsartan 97/103 2 tablets; amlodipine 5 mg 2 tablets; luvion 25 mg 1 tablet; clonidine patch 0.1mg; acetylsalicylic acid 200 mg 1 tablet; coumadin according to INR values and pain relievers as needed. The mean of the patient’s systolic blood pressure, measurable by the oscilloscope, was between 85 and 90 mmHg; body weight was 72 kg, pulse oximeter saturation was 97%, and there was a rhythmic heart rate of 73 beats per minute. The electrocardiogram showed normal atrioventricular conduction, left anterior hemiblock, outcomes of extended anterior necrosis with hemocyte-negative T; QTc 439 milliseconds. There was reduced vesicular breath sounds at the pulmonary bases with no pleural effusions (Figure 1).

The patient was diagnosed with obstructive sleep apnea syndrome (OSA), with an apnea-hypopnea index (AHI) of 15–30 (moderate OSA), and with indications for nocturnal continuous positive airway pressure (CPAP). During previous hospitalization (for the LVAD implant), the patient complained of the presence of non-specific dull pains in the left facial area, left shoulder, and neck, as well as deep pain in the lingual area (at the base). He complained of odynophagia and an increase in the symptoms described during active movement of the neck and jaw (including swallowing). This symptomatology became more pronounced in the first days of the second hospitalization, i.e., in the cardiopneumology department. The administration of the visual analog scale (VAS, 0–10 cm) showed a value of 6. Excluding purely cardiac reasons, from positioning of the LVAD to a possible myocardial infarction, it is not easy to identify the cause of pain in patients with an LVAD [14]. Through instrumental tests, we excluded joint problems in the shoulder and cervical tract due to the constant transport of the accessory bag (for batteries), and eliminated possible causes from the gastroesophageal area, as well as disorders from the central nervous system. In order to obtain some preliminary information on the function of the diaphragm muscle, we had the patient perform the Bordoni Diaphragmatic test (BDT), which was positive; BDT remained positive even after OMM treatment [15]. In patients with LVAD implantation, there is a reduction in the movement of the left anterior hemi-diaphragm, for various post-surgical reasons and due to the presence of the machinery, and the pulmonary/diaphragm function does not improve over time [16,17].

Palpation of the cervical area revealed that, at the level of C3–C4, the tissues were painful and stiff, with decreased range of active and passive movement; the joints that allow the complex movement of the shoulder (clavicle, scapula, humerus) did not show particular motor and tissue anomalies. The temporomandibular joint, albeit slightly limited in the opening to the left, did not show abnormal joint noises or dysfunctions. The active, open mouth protruding movement of the tongue appeared limited (and with pain stimulation), with no observed deviations; the morphology and color were normal. Placing a finger in front of the tip of the tongue to test lingual strength during protrusion, the pain reported by the patient, at the base of the tongue on the left, increased.

We performed a non-invasive and non-instrumental test for the lingual complex (Performance Tongue Test—PTT), in order to obtain preliminary indications of the lingual function, which was positive [18]. The buccal floor was generally tense and slightly painful, but without a specific location of a painful area. The palpatory evaluation was performed with the patient supine and, with the same evaluation procedure, the palpation was performed with the patient seated. Finally, laterally and manually translating the great horns of the hyoid bone, the patient complained of localized pain (greater horn of the left), accentuated when the extremity of the hyoid bone was brought to the left, with radiation to the left shoulder, the cervical tract, and the left facial area.

The nociceptive-type reaction from translating the hyoid bone to the left, with localized and radiating pain, allowed us to frame the patient as suffering from hyoid bone syndrome (HBS), which will be discussed in the next section. HBS is caused by inflammation of the attachment of the stylohyoid ligament and/or of the middle pharyngeal constrictor to the greater horn of the hyoid bone [19,20]. The patient underwent OMM for three sessions of a few minutes each (maximum 120 s), over a week. The approach chosen was the application of unwinding, taking the tongue (with gauze and gloved hand), while the other hand held the great horns of the hyoid bone between index and thumb; the patient remained in the supine position. The anterior portion of the tongue (retrusive attitude at rest) was slightly pulled forward, while the hand on the hyoid bone was listening (passive).

Once a slight lingual tension was created, the osteopath waited for the tissue to freely express chaotic movements, and then passively followed its course; the task of the listening hand was to verify that the hyoid did not undergo movements imposed by the tongue. The technique ends when the osteopath’s hand feels that the movements of the tongue are balanced, with respect to retrusion and protrusion actions; the technique ends earlier if the patient feels pain or discomfort. The hand holding the tongue is in movement, but it is a movement that reflects that expressed by the tongue. Initially, the movement of the lingual complex is imperceptible, until preferential movement is expressed, where the action of the tongue is more visible. It is not always possible to reproduce the exact manual approach with different patients, as the unwinding bases the application time on the response that the tissue or anatomical area of the same patient expresses. With this technique, it is the clinician who adapts to the patient and not the patient who adapts to the clinician (Figure 2).

The week after the last of the three weekly sessions, the same palpatory evaluation, with the patient supine and then sitting, revealed the disappearance of the stiffness in the cervical tract, with full range of motion; nothing relevant emerged from the evaluation of the shoulder. The buccal floor showed no stiffness and the odynophagia disorder disappeared. The movement of the jaw did not yield any pain, compared to the previous evaluation. The manual translation of the hyoid bone to the left did not trigger any symptoms, either local or radiating. The assessment of pain perceived in the hyoid area, through the administration of the VAS, was equal to one. At one and three months after discharge, during the general cardiovascular evaluation visits, these palpatory evaluations were repeated; the result of the VAS was then zero.

## 3. Discussion

The presented clinical case describes the resolution of pain deriving from HBS, through an osteopathic approach (lingual unwinding), in a patient with an LVAD. HBS, or insertion tendinitis or tendinosis of the muscle–tendon area of the great hyoid horn, was reported for the first time by Brown in 1954 [19]. This syndrome is likely the response of previous inflammation due to repeated small trauma of the cervicopharyngeal area, or a direct trauma to the hyoid area involving the soft tissues of the great horn. At the histological level, tissue fibrosis, scar tissue, atrophic contractile tissue, and the presence of calcifications can be found [21].

The classification of HBS, is not immediate and the clinician is not accustomed to taking this into consideration [19]. Symptoms of HBS, which can be confused by a precise localization of the symptomatic genesis, can be linked to dysphonia (due to alterations in the perilaryngeal musculature), temporomandibular and odontogenic pain, glossopharyngeal neuralgia, Eagle’s syndrome, craniofacial neuralgia and retropharyngeal calcific tendinitis, esophageal diverticula, and a presence of tumoral formations [21,22]. A discriminating evaluation to shed light on the causes of pain is the palpation and active movement of the patient’s hyoid bone by the clinician; the appearance of pain in the area corresponding to the great hyoid horn is a diagnostic test [19,20,22]. Instrumental examinations are not necessarily considered a diagnostic gold standard [19]. The usual therapeutic approach can be surgical (hemihyoidectomy), when the symptoms are refractory to any treatment [20]. The clinician can perform local injections (one injection has a 70% resolution success) of cortisone or local anesthetics; generally, this approach has a high success rate (80.98%) [21,22]. In the case of confirmed HBS, the clinician can administer nonsteroidal anti-inflammatory drugs by mouth or using topical applications, but the resolution rate drops to 61% [21]. Other therapeutic solutions involve physiotherapy and speech therapy, utilizing massages and exercises to improve posture [19,22].

The hyoid bone is always in motion, through the musculature that is inserted above and below its surface, during the acts of breathing, swallowing, chewing, and speaking [19,22]. To give an example, during swallowing, the hyoid bone is pulled anteriorly and superiorly (10.9 ± 2.8 mm and 8.3 ± 4.1 mm, respectively) by contraction of the suprahyoidal muscles and the hyoglossus muscle [23]. It is reasonable to think that the muscle–tendon system involved with the hyoid bone can undergo degeneration processes over time, or due to the presence of direct or indirect trauma (such as whiplash), triggering the symptoms [20].

The retrusive attitude of the tongue in the patient with the LVAD may be linked to the presence of obstructive sleep apnea syndrome, in which a lower position of the hyoid, with a retrusive tongue attitude has been demonstrated [24]. We do not know the exact causes that led to the presence of HBS in the patient; likely, in addition to a possible caudal position of the hyoid (causing excessive tension on the great horn, superiorly), the symptoms could have arisen from a muscular imbalance of the hyoid bone area due to sarcopenia. Patients with a long history of chronic heart failure may have a loss of muscle mass in general, and only after LVAD implantation can they recover the volumes lost in the first six months [25].

The use of the unwinding technique derives from the fact that, in the literature, it has been used successfully to relieve pain in cardiac surgery patients [26,27,28]. The use of this technique on the tongue finds its basis in the literature, where texts with the unwinding approach and positive results appear, although there is a total absence of the treatment of the tongue in the reference book of the OMM (Foundations of osteopathic medicine: philosophy, science, clinical applications, and research, 4th ed. Philadelphia, PA: Wolters Kluwer; 2018) [26,29]. Working the tongue means working on the tensional balance of the hyoid area, thanks to the preferential relationship with the hyoid.

The articles discussing HBS do not report an explanation of the presence of pain radiating to other areas, which may be distant from the hyoid bone.

The middle pharyngeal constrictor muscle is managed by a central pattern generator (CPG), through complex neural connections between the spinal cord and the brain stem; CPG controls the functions of all peri-hyoid muscles [30]. The glossopharyngeal nerve (IX cranial nerve) innervates the middle pharyngeal constrictor muscle and sends parasympathetic fibers to the stylohyoid ligament [31]. The IX anastomoses with the lingual nerve (V cranial nerve), with the vagus nerve (X cranial nerve), and with fibers of the sympathetic upper cervical ganglion; anastomoses create the pharyngeal plexus (PP), behind the middle pharyngeal constrictor muscle [31]. The IX also anastomoses with other fibers, with the facial nerve (VII), and the hypoglossal nerve (XII cranial nerve) [31]. The X innervates a portion of the diaphragm muscle and must act in perfect synergy with the phrenic nerve [18].

We can hypothesize that persistent dysfunction of the diaphragm muscle could alter the function of the vagus nerve, negatively affecting the functionality of the pharyngeal area and tongue, and creating abnormal tensions suffered by the hyoid bone [30]. Muscular and ligamentous tensions of the hyoid bone could create non-physiological mechano-metabolic conditions, and this information could be transported in a nociceptive manner [32]. Finally, the anastomoses of the X and IX could explain the facial symptoms, as well as those of the shoulder (the vagus nerve anastomoses with the brachial plexus) and neck (the XI anastomoses with the X), the odynophagia, and the pain in the tongue. The positivity of the BDT can be explained by the fact that the diaphragm, in an LVAD patient, remains in dysfunction; the negativity of post-OMM PTT can be explained by the improvement in the muscle–ligamentous tensions of the hyoid area, with optimal lingual function. The patient’s symptomatic improvement, following the application of the osteopathic approach, could be explained by the fact that this technique (similar to the gentle manual techniques) stimulated the intervention of the vagus nerve [33]. We know that the intervention of the X cranial nerve has a positive effect on the inflammatory environment [34]. Compared to invasive treatment for the resolution of the disorder, the manual approach has no side effects and is well tolerated by the patient. The local and systemic response of the manual unwinding technique is not known. We do not yet have data to understand which specific adaptations occur. Likely, the gentle approach carried out produces a mechanotransductive response capable of positively influencing the peripheral and central nervous system [35].

### Future Directions and Clinical Implications

Future Directions and Clinical Implications

We need more information to understand the causes that lead to the symptoms of HBS, as well as to encourage the clinical evaluation of the presence of this syndrome, e.g., with the palpation of the hyoid bone. Researchers should make more efforts to fully elucidate the causes of this syndrome. HBS can cause psychological distress and fear when swallowing, which could influence the patient’s eating behavior. It would be interesting from a clinical point of view if, as a habit, speech therapists could assess the hyoid area after surgery. In this way, dysfunctions could be intercepted more quickly, and the patient’s discomfort time could be reduced.

## 4. Conclusions

The case report describes the presence of hyoid bone syndrome (HBS) in a patient with a previous history of chronic heart failure and subsequent implantation of a left ventricular assist device (LVAD). HBS is not easy to identify clinically, as the painful symptoms it triggers are both local and distant, with respect to the anatomical area of the hyoid. Palpation by the osteopathic clinician made it possible to highlight HBS as a symptomatic cause and to resolve, through an unwinding-type technique applied to the tongue, the patient’s painful disorders, related mainly to pain in the neck and face, as well as with swallowing. We do not know, in detail, the reasons that lead to HBS, and more research should be directed towards this syndrome.

## Figures and Tables

**Figure 1 healthcare-11-01130-f001:**
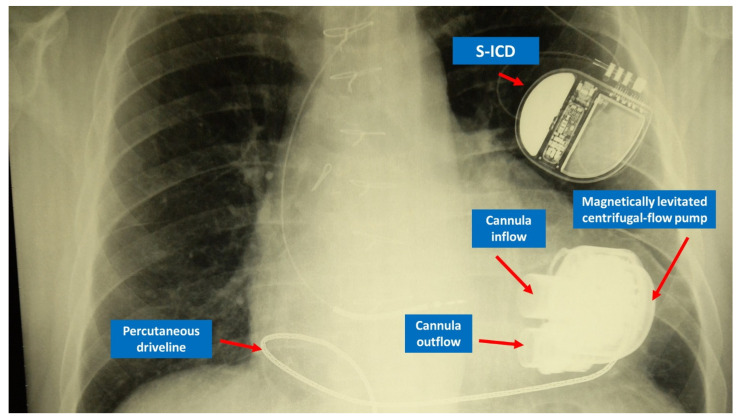
The X-ray shows the presence of S-ICD, LVAD (cannula inflow at the apex of the left ventricle, cannula inflow and outflow, magnetically levitated centrifugal-flow pump), and the percutaneous driveline for external batteries. S-ICD: subcutaneous implantable cardioverter defibrillator; LVAD: left ventricular assist device.

**Figure 2 healthcare-11-01130-f002:**
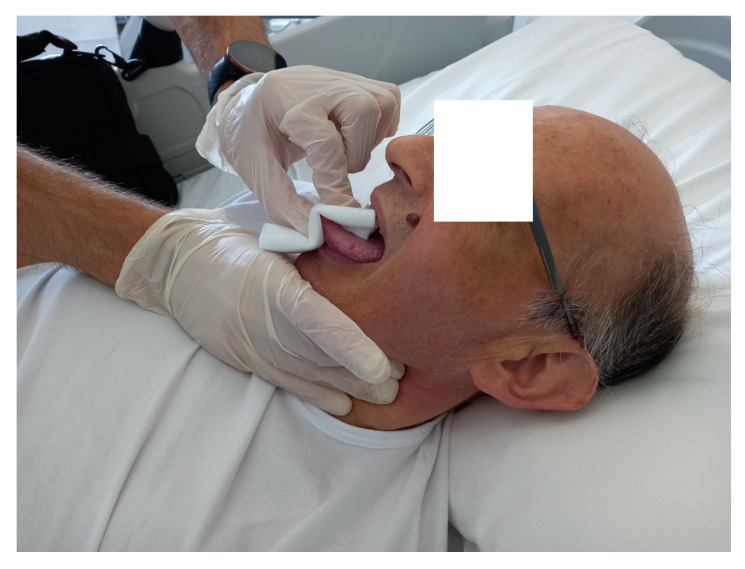
The patient is supine, with the osteopath’s caudal hand near the great horns of the hyoid bone, while with the cranial hand, he holds the tip of the patient’s tongue, which is slightly protruded, with gauze. The first hand is listening, while the second hand performs the unwinding technique.

## Data Availability

The ethics approval is not required for case report.

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
