# Peer review of "Hyoid Bone Syndrome in a Patient Undergoing Left Ventricular Assist Device Implantation"

_healthcare, 2023, doi:10.3390/healthcare11081130_

Round 1
Reviewer 1 Report
This case report describes the resolution of an oropharyngeal problem, caused by the presence of hyoid bone syndrome (HBS) in a patient with LVAD, through osteopathic manipulative medicine (OMM) of the tongue. To the knowledge of the authors, this is the first clinical case in the literature where HBS, OMM, and a patient with HM3 are highlighted.
The authors can hypothesize that persistent dysfunction of the diaphragm muscle could alter the function of the vagus nerve, negatively affecting the functionality of the pharyngeal area and tongue, and creating abnormal tensions suffered by the hyoid bone. Muscular and ligamentous tensions of the hyoid bone could create non-physiological mechano-metabolic conditions, which information could be transported as nociceptive.
The positivity of the BDT can be explained by the fact that the diaphragm, in an LVAD patient, remains dysfunctional; the negativity of post-OMM PTT can be explained by the fact that an improvement in the muscle-ligamentous tensions of the hyoid area are improved, with an optimal lingual function. The patient's symptomatic improvement following the application of the osteopathic approach could be explained by the fact that this technique (like the gentle manual techniques) stimulated the intervention of the vagus nerve. We need more information to understand the causes that lead to the symptoms of HSB, as well as the clinical habit of taking into consideration the possibility of the presence of this syndrome, for example with the palpation of the hyoid bone.
The article is well written and interesting to read, as it is unusual to see such a complication in patients requiring LVAD for left ventricular dysfunction.
The level of scientific English is sufficient for good comprehension, but I recommend a full and thorough review.
Author Response
Dear Reviewer 1,
Thanks for taking the time to review and comment.
The text has been revised by my colleague Escher, who is American. Maybe it changes a bit with European English. We believe that the text is correct, but if the Editor requests a revision of the language, we will provide it.
Reviewer 2 Report
The authors have developed a well-conducted and well-written case report with the aim of analyzing the presence of a hyoid bone syndrome (HBS) in a patient with left ventricular assist device (LVAD) implantation, and the resolution of painful symptoms through an osteopathic manual technique (unwinding) applied to the tongue
However, I suggest some clarifications or modifications that will in my opinion improve the quality of their manuscript:
1. I recommend the authors to add the type of study in the title.
2. In the Introduction/Discussion section, I recommend that the authors comment on a recent preliminary study describing certain global osteopathic techniques, which could also have been used in their patient, mentioning the following reference: DOI: 10.3390/ijerph20021061
3. In the Discussion section, I recommend that the authors compare their results and comment by discussing with a recent review comparing the effects of HIIT and MICT on aerobic fitness and quality of life in patients undergoing coronary artery bypass graft surgery , citing the following reference: DOI: 10.3390/jcdd9100328
4. In the Discussion section, could you add a section on "Future Directions and Clinical Implications"?
Author Response
Dear Reviewer 2,
Thanks for taking the time to review and comment.
I recommend the authors to add the type of study in the title. We have added "case reports".
In the Introduction/Discussion section, I recommend that the authors comment on a recent preliminary study describing certain global osteopathic techniques, which could also have been used in their patient, mentioning the following reference: DOI: 10.3390/ijerph20021061
Thanks for the suggestion, but the manual technique applied is unwinding and not the manual approaches described in the suggested article (Maitland’s technique and low load and long duration to the myofascial complex until the tissue restriction disappears).
In the Discussion section, I recommend that the authors compare their results and comment by discussing with a recent review comparing the effects of HIIT and MICT on aerobic fitness and quality of life in patients undergoing coronary artery bypass graft surgery , citing the following reference: DOI: 10.3390/jcdd9100328
We believe that the theme of the suggested article (where the same author still appears) is not relevant to the theme of the case report.
In the Discussion section, could you add a section on "Future Directions and Clinical Implications"?
We have added this paragraph “Future Directions and Clinical Implications” to the end of the "Discussion" paragraph and added some sentences (Research should make more efforts to fully elucidate the causes of this syndrome. HBS can cause psychological distress and fear when swallowing, which conditions could influence the patient's eating behavior. It would be interesting from a clinical point of view if, as a habit, speech therapists could assess the hyoid area after surgery. In this way, dysfunctions could be intercepted more quickly and the patient's discomfort time reduced.)
Reviewer 3 Report
The describe case is interesting. I would ask for more details on how the procedure is administered (e.g. exact duration, precise techniques used) to maximize the standardization and the reproducibility of the method, mostly because, as stated by authors, it is quite innovative.
Author Response
Dear Reviewer 3,
Thanks for taking the time to review and comment.
The describe case is interesting. I would ask for more details on how the procedure is administered (e.g. exact duration, precise techniques used) to maximize the standardization and the reproducibility of the method, mostly because, as stated by authors, it is quite innovative.
In the "Case description" section we added "(maximum 120 seconds)"
The technique used, as written and described in the text of the same section, is the unwinding.
We have added “The hand holding the tongue is in movement, but it is a movement that reflects that expressed by the tongue. Initially, the movement of the lingual complex is imperceptible, until the preferential movement is expressed, where the action of the tongue is more visible. It is not always possible to reproduce the exact manual approach with different patients, as the unwinding bases the application time on the response that the tissue or anatomical area of the same patient expresses. With this technique it is the clinician who adapts to the patient and not the patient who adapts to the clinician.”
Reviewer 4 Report
Abstract
The context needs to be modified by referring to the CARE guidelines.
Introduction
Contents should be described focusing on hyoid bone syndrome.
The description of LVAD is long. Consider reducing content.
Please focus your description on the symptoms and signs of HBS.
Case description
The context needs to be modified by referring to the CARE guidelines.
For systematic reporting, it is considered that it is better for readability to classify and describe each subheading.
Discussion
Describe your strengths compared to conventional treatments
Interpretation of the pathogenesis of HBS is also important, but detailed information on interventions is needed.
Author Response
Dear Reviewer 4,
Thanks for taking the time to review and comment.
Abstract
The context needs to be modified by referring to the CARE guidelines.
We have followed the recommendations of the Healthcare Journal.
We have read the suggested article (J Med Case Rep. 2013 Sep 10;7:223. doi: 10.1186/1752-1947-7-223). The indications of the CARE guidelines are subjective and are not mandatory; the authors themselves write: “The CARE guidelines and their development have several possible limitations. First, these guidelines were developed through a consensus method and thus represent the opinions of the participants”
Introduction
Contents should be described focusing on hyoid bone syndrome.
The syndrome is described in the “Discussion” section. We believe that the organization of the text can meet the needs of the reader.
The description of LVAD is long. Consider reducing content.
We believe that the reader does not know the topic related to the LVAD and that it is useful to have a global clinical vision.
Please focus your description on the symptoms and signs of HBS.
Unfortunately, there are not many texts on the subject in the literature and everything we have written reflects current research. The syndrome is described in the “Discussion” section.
Case description
The context needs to be modified by referring to the CARE guidelines.
We have followed the recommendations of the Healthcare Journal.
For systematic reporting, it is considered that it is better for readability to classify and describe each subheading.
We have followed the recommendations of the Healthcare Journal.
Discussion
Describe your strengths compared to conventional treatments
We added at the end of the "Discussion" paragraph: Compared to an invasive treatment for the resolution of the disorder, the manual approach has no side effects and is well tolerated by the patient.
Interpretation of the pathogenesis of HBS is also important, but detailed information on interventions is needed.
We added at the end of the "Discussion" paragraph: The local and systemic response of the manual unwinding technique is not known. We do not yet have data to understand which specific adaptations occur. Probably, the gentle approach carried out produces a mechanotransductive response capable of positively influencing the peripheral and central nervous system [35].”
Round 2
Reviewer 3 Report
Authors have completely answered my comments. I endorse the publication.
Reviewer 4 Report
The changes made by the authors are satisfactory.